# Influence of Environmental Aesthetic Value and Anticipated Emotion on Pro-Environmental Behavior: An ERP Study

**DOI:** 10.3390/ijerph19095714

**Published:** 2022-05-07

**Authors:** Huie Li, Chang You, Jin Li, Mei Li, Min Tan, Guanfei Zhang, Yiping Zhong

**Affiliations:** 1Department of Psychology, Hunan Normal University, Changsha 410081, China; lihuie@hunnu.edu.cn (H.L.); changyoupsy@hotmail.com (C.Y.); jimli09@163.com (J.L.); 18352265205@163.com (M.L.); mintan2021@163.com (M.T.); 18364915905@163.com (G.Z.); 2Cognition and Human Behavior Key Laboratory of Hunan Province, Changsha 410081, China; 3Department of Psychology, School of Education Science, Hunan Normal University, No. 36 Lushan Road, Changsha 410081, China

**Keywords:** emotion regulation, anticipated guilt, anticipated pride, N1, P2, N2, LPP

## Abstract

Perceptual cues act as signals in the aesthetic value environment, which promote emotion regulation toward pro-environment behavior. This type of perception-emotion-behavior reactivity forms the core of human altruism. However, differences in pro-environmental behavior may result from variation across high-aesthetic-value (HAV) and low-aesthetic-value (LAV) environments. This study investigated the neural mechanisms underlying interaction effects between environmental context and emotion regulation on pro-environmental behavior by integrating behavioral and temporal dynamics of decision-making information processing with event-related potential (ERP) technique measures. The results indicated that changing anticipated emotions changes pro-environmental behavior. Regarding changing aesthetic value environments, while modulating emotion regulation, significant differences were found in brain regions and mean amplitudes of N1, P2, N2, and late positive potential (LPP) components, which anticipated emotion. The findings suggest that environmental aesthetic value and emotion regulation impact pro-environmental behavior.

## 1. Introduction

People from diverse backgrounds may often experience the same event differently, which results in divergent behaviors [1,2,3,4,5,6]. In other words, multiple perspectives can result in a diverse understanding, and thus, varied behaviors. In many studies, this phenomenon is described as the framing effect [7,8,9,10]. At present, environmental protection is one of the most important concerns for human existence [11]. The motivations for it are specifically based on two opposing aspects. First, regarding a high-aesthetic-value (HAV) environment, to ensure that the environment continues to be suitable for human survival, actions toward its protection should be prioritized. Second, the environment needs to be protected well because human beings cannot survive in polluted and destroyed environments. Therefore, it is important to repair the living environment. The final behavior resulting from these two antecedents is an individual’s pro-environmental behavior.

Several studies have been conducted on pro-environmental behavior. It refers to behavior that induces little to no harm or that benefits the environment [12]. It encompasses every aspect of life, and everyone can engage in pro-environmental behaviors as best as they can [13,14,15,16,17,18].

According to the perception-emotion-behavior model, the reasons for human engagement in pro-environmental behaviors may be motivated by external environmental cues or by internal motivations. Furthermore, the physical environment wherein humans engage in pro-environmental behavior is the immediate external factor that plays an important role in their behavior [17]. Along with the aesthetic impressions people receive from their environment [19], a wide range of psychophysiological responses are affected by it [20], including memory [21], stress [22], attention [23], and well-being [24]. Studies have suggested that participants with a stronger intention of engaging in moral behavior in an environment were associated with having a HAV, whereas a decreased intention was associated with a low aesthetic value (LAV). Thus, an environment with a LAV is related to immoral behavior [25]. Pro-environmental behavior is considered altruistic and moral behavior. Studies have found that pictures of a positive emotional environment can promote positive emotions and are associated with a reduced intention to take action to protect the environment. In contrast, pictures of negative emotional environments augmented negative emotions and increased the willingness to pay attention to and protect the environment [26].

The perception-emotion-behavior response is the core of human altruism, which frequently occurs in real life [27]. Scholars have indicated that individuals experience positive or negative emotions in their daily life and can anticipate their emotion from engaging in a particular behavior [11]. Anticipated emotions are based on future outcomes [11]. Hence, the emotions people experience while in the physical environment will be similar to the imagined future environment. The emotions that they have not experienced will significantly impact the individual’s cognitive processes [28], behaviors [29], and decision-making processes [30], the same as the experienced emotion. Anticipated feelings, such as pride and guilt, are often regarded as self-conscious emotions. The self-conscious emotion is a special type of emotion that involves the ego, and it is particularly relevant to comprehending environmentally responsible decision-making processes or behaviors consistent with the Norms Activation Model (NAM) [31]. Furthermore, these feelings have a distinct role in an individual’s decision-making processes for pro-social and -environmental behavior [11,28,32]. Self-conscious emotions have self-regulatory functions. Onwezen and colleagues confirmed that anticipated emotions mediate the effects of personal norms on behavior and affected behavior through behavioral intentions [11]. Moreover, anticipated emotions are often overestimated such that the emotions one anticipates are more extreme than the ones experienced during the event [30]. Individuals strive to experience positive emotions and avoid negative ones. Research findings imply that there was an increased sense of environmental moral responsibility among individuals who felt guilty about past or failed actions [33]. Anticipated emotions are particularly relevant in understanding individual decision-making [34,35,36,37,38,39].

Scholars have described the anticipated emotional response based on the prospect of success and failure as pre-evaluation, also known as pre-thinking or anticipated thinking [40]. Emotions emerging from these personal or social standards promote prosocial or altruistic behavior [41] and have been examined for their influence on pro-environmental behavior [41]. However, factor analysis revealed inconsistencies in individual environmental behavior [12]. For example, people might exhibit environmentally friendly behaviors in waste recycling but environmental burdens regarding transportation [42]. Moreover, research has suggested that different types of information could either promote or dampen pro-environmental behavior and subsequent spillover effects [43]. Another finding regarding the strong support for environmental protection was contrasted by low actual pro-environmental behavior [44].

Regarding inconsistencies between behavioral decisions and situations, the framing effect is worth mentioning. One of the most robust and important findings in the psychology of decision-making is the dramatic influence of framing [45,46]. At times, framing is simply choosing the correct way of labeling an environmental policy that can influence choice. However, in a much broader sense, it can have implications for the way people view environmental issues and the extent to which they feel morally responsible and motivated to engage in pro-environmental behavior [47].

Research on the effects of pride and guilt on message framing found that pride and guilt have different behavioral effects when combined with positive and negative framing, respectively [48,49,50,51]. The event-related potential (ERP) is a neuroscience technology with a high temporal resolution property, which can be used to investigate the temporal dynamics of decision-making information processing [52,53,54]. ERP research on message framing regarding green purchase established that the influence of positive framing was higher on participants’ intention, which was supported by different ERP components, N1\P2\late positive potential (LPP) in the three stages of time processing. The effects of attentional resources, guilt, and pride impact purchase intention when processing positive and negative message framing [55]. Specifically, the study found that increased pride and decreased guilt through positive framing can increase purchase intention for green products, whereas increased guilt and decreased pride can reduce it through negative framing. This finding differs from previous views that positive emotions reduce the intention to protect the environment, and negative emotions increase the willingness to pay attention and protect the environment [26]. Subtle positive emotional environments that are not task-related could significantly impact decision-making.

Evidence revealed that environmental pictures of positive emotions presented before an individual’s decision were sufficient to neutralize the framing effect by reducing the aversion aspect of affirmative loss [46]. While the participants regulated their emotions, a consistent trend was observed for the negative conditions wherein pictures caused increased regulation to elicit the most positive LPP amplitude, while maintaining the negative emotion condition elicited a more minimal LPP amplitude [56]. The study results showed a significant effect of the regulation condition for a time window of 350 to 750 ms for the LPP amplitude occurring at 350 to 3000 ms [56]. Furthermore, research demonstrated that the LPP generation, which Ochsner and Gross (2007) referred to as the perceptual appraisal system, is attributed to the parietal, occipital, and temporal cortices [57]. Moreover, the decreasing emotions may incorporate and exacerbate memory interpretations associated with the LPP modulations during decreased instructions because its magnitude reflects activity [57]. To support changes in emotional experience, studies recommend that alternative stimulus representations be generated in the perceptual appraisal system during emotion regulation [58].

Two hypotheses were tested in this study: First, there is a difference between the environmental behavior of individuals in HAV and LAV environments, with the LAV environment expected to present more pro-environmental behavior. Second, in the process of increasing and decreasing anticipated emotions through environmental framing, neural processes are associated with individuals’ environmental behaviors, and the ERP components are changed by different tendencies under the circumstance of the HAV and LAV. We designed a 2 (environmental framing: negative vs. positive) × 2 (anticipated emotion: pride vs. guilt) × 2 (group: HAV vs. LAV) mixed experimental design to explore how environmental aesthetic values and anticipated emotions influence pro-environmental behavior through environmental framing. In this study, pro-environmental behavior refers to the specific amount that participants allocate to environmental organizations during the course of the experiment in a monetary allocation task related to their own reward for participating in the experiment.

## 2. Materials and Methods

### 2.1. Participants

For this study, 73 Chinese college students including 41 women, aged 18 to 24 years (*M* = 20 years, *SD* = 1.15 years), were recruited from Hunan Normal University and randomly allocated to two groups. The 37 participants assigned to the HAV group viewed HAV environmental pictures in the HAV condition, and the 36 participants assigned to the LAV group viewed the LAV environmental pictures in the LAV condition [20]. The participants did not have any neurological or psychiatric diseases, and their vision was either normal or corrected to normal. The study protocol was approved by the Ethics Committee of the Management Laboratory of Hunan Normal University. All experiments were performed following relevant guidelines and regulations and all the participants provided written informed consent before the experiment.

### 2.2. Stimuli and Procedure

#### 2.2.1. Stimulus Materials

A material evaluation of the environment was performed, and 64 environmental pictures were selected. Half of them portrayed well-preserved natural environments, and the remaining 32 portrayed polluted environments (including air and water pollution) with obvious traces of human activities, without environmental protection. All the pictures were obtained from the Baidu Image Library. Adobe Photoshop (San Jose, CA, USA) was used to perform unified standard post-processing on the pictures, with a unified resolution of 1440 × 900 pixels and a size of 500 × 333 pixels. Material evaluation data were collected using Wenjuanxing, which is a professional online questionnaire platform in China. Then, 49 college students were recruited to evaluate the photos from the six dimensions of the three aesthetic models [20]. After each photo was presented, seven sliding bar scoring questions were asked in a 7-point Likert scale assessment (1 = low anchor, 7 = high anchor) (see Table 1).

For the material evaluation, 49 valid questionnaires were collected (45 girls, 18–25 years, *M*_age_ = 19.43, *SD* = 1.37). The results revealed significant differences among the environmental pictures. Specifically, the variety was due to the order, beauty, comfort, valence, and relaxation with HAV and LAV. Furthermore, there was no significant difference in complexity, meaning that the image structure had no difference in the quality gap. LAV environment pictures were associated with a higher level of depression and uncomfortable emotions and experiences (Table 2). For this study, 20 pictures from both types of study material were screened.

#### 2.2.2. Procedure

Each participant received instructions and information about the experiment’s process, followed by signing the informed consent form before participation. When introducing the experimental process, we first presented the scientific research projects between the environmental protection organizations and our laboratory. Our study was supported by this research project. The research task was to examine people’s sensitivity to money (in order to obscure the experimental aim). The specific task was to allocate money, with each distribution related to participants’ reward and a public welfare fund raised by an environmental protection organization (the China Environmental Protection Agency (CEPA)). Participants were informed that they would allocate the money many times and that each time they were given CNY 10 (RMB), they owned that money. They decided their own share of the CNY 10, that is, they could decide how much to give to CEPA, and the rest they could keep for themselves. To ensure the ecological validity of the experiment, we informed the participants of the following: we randomly selected one block (there were four blocks in the experiment and 60 trials in each block) and averaged the number of allocation trials after completion of the task as their reward. In other words, the participants’ allocations were directly related to their compensation for participating in the experiment.

There were 240 trials in the formal experiment, and four practice trials were conducted to familiarize participants with the experimental procedure. The experiment was designed using four unrepeated blocks and was run using the E-prime 3.0 software package (Psychology Software Tools, Pittsburgh, PA, USA). It was conducted in an electrically shielded room with dim lighting. There was a 100 cm distance between each participant and the computer screens. Each participant received written instructions to complete the experiment and used a keyboard to record their choices. In each trial, they viewed a fixation point for 1000 ms, then an environmental picture for 1500 ms, followed by an environmental framing message for 2000 ms. Finally, they input the amount of money that they wanted to allocate to CEPA from the CNY 10 every trial. They could enter the next trial after they made their response (press key to enter numbers). Environmental framing was used to alter the anticipated emotion, namely, “I feel pride about protecting the environment” and “I would not feel pride about polluting the environment” to increase and decrease anticipated pride, respectively, as performed for anticipated guilt. Based on the information provided in the message, participants were asked to decide whether they wanted to allocate the amount for CEPA. The participants recorded their choices using numbers ranging from 1 to 10 on the keyboard, representing the amount of money allocated to CEPA (see Figure 1). On average, every participant was rewarded with CNY 50 to 70.

#### 2.2.3. Electrophysiological Recording

The electroencephalogram (EEG) data were recorded (band-pass filter of 0.01–30 Hz, sampling rate of 500 Hz) with a Bran Product (Brain Products, Gilching, Germany). Data were recorded on 32 scalp sites using Ag/AgCl electrodes for the standard international 10 to 20 system. Electroencephalograms were recorded from electrodes located near the outer canthus of each eye (horizontal) and above and below the left eye (vertical). During the experiment, the electrode impedance was maintained below 5 kΩ. Fz was used as the reference electrode. Offline, EEG data were calculated as the average of the left and right mastoids. Digital filtering of electroencephalogram artifacts was performed through a zero-phase shift (high-pass at 30 Hz, 24 dB/octave). The EEG level for epochs was set between −200 and 1000 ms. Baseline correction was made using a 200 ms interval pre-stimulus onset. Artifacts beyond ±70 μV were rejected.

Emotional stimulations were required to elicit additional perceptual processing and related stimuli for the intrinsic task-related effects of N2 and LPP. Previous ERP studies showed that emotional responses were characterized by increased LPP amplitude. Based on previous research and combined brain topography, ERPs for emotion regulation and anticipated emotion were characterized by an early negative activity at 130 to 190 ms (N1). The positive activity was 200 to 280 ms (P2) in the central/back regions, which were followed by one negative activity at 270 to 330 ms (N2) and a positive activity at 400 to 800 ms (LPP). To avoid potentially significantly false effects on ERP amplitudes due to multiple comparisons, the mean values of the amplitudes of the N1, P2, N2, and LPP components were calculated at central-posterior electrodes (i.e., C3/CZ/C4/CP5/CP6/CP1/CP2/P3/PZ/P4) [59]. The components that emerged during the temporal processing of the 10 electrode points were selected to analyze how HAV and LAV environmental and emotion regulation interacted, affecting the individual’s pro-environmental behaviors.

## 3. Results

### 3.1. Behavior Data Analysis

#### 3.1.1. Result of the Allocation Amount 

Repeated-measures analyses of variance (ANOVAs) were conducted to determine the mean allocation amount for CEPA. The independent within-subjects variables were anticipated emotion (anticipated pride vs. guilt) and emotion regulation (increasing emotion vs. decreasing emotion). The HAV and LAV environment groups were identified as between-subjects variables. As expected, the results revealed a significant effect on the main findings of the anticipated emotion, *F*(1,71) = 12.33, *p* = 0.001, η_p_^2^ = 0.15 and the emotion regulation, *F*(1,71) = 24.3, *p* < 0.001, η_p_^2^ = 0.26. Furthermore, the interaction effect between emotion regulation and anticipated emotions was significant, *F*(1,71) = 6.68, *p* = 0.012, η_p_^2^ = 0.086. A further simple effect analysis indicated that when participants received the message of increasing anticipated pride and guilt, they would allocate more money to CEPA, and when they received the message of decreasing anticipated pride and guilt, they would allocate less money to CEPA. The trend of allocation was consistent with the direction of emotion regulation. Specifically, with increased participants’ pride, the mean allocating amount (*M* = 5.7, *SE* = 0.28) exceeded the increased guilt (*M* = 5.39, *SE* = 0.28), with a significant difference of *F*(1,71) = 11.44, *p* = 0.001, η_p_^2^ = 0.14. However, when participants received the message of decreasing anticipated pride, the mean allocating amount (*M* = 5.21, *SE* = 0.28) was less than for decreasing guilt (*M* = 5.25, *SE* = 0.28), and the difference was not significant (see Figure 2 and Table 3).

The behavioral results suggested there was no significant difference in the allocation amount to CEPA between HAV and LAV environment groups, *p* = 0.34. Under the circumstance of the interacted effect of emotion regulation and aesthetic value environment, the individual’s pro-environmental behaviors showed no difference regarding HAV and LAV environments.

#### 3.1.2. Result of Response Time

The results of a repeated-measures ANOVA showed that there was no significant difference in response time within the study of 2(emotion regulation: increase vs. decrease) × 2(aesthetic value environment: high vs. low). The findings suggested that response time was not affected by the environmental message framing and aesthetic value environment.

### 3.2. ERP Results

An ANOVA of the average amplitudes of N1, P2, N2, and LPP components was conducted with anticipated emotion (AP vs. AG), emotion regulation (increase vs. decrease), and hemisphere (right, midline, and left) as within-subjects factors, and the HAV and LAV groups were between-subjects variables. Post hoc testing of the significant main effects was conducted using Bonferroni’s method. *p*-values were corrected by Greenhouse–Geisser correction.

#### 3.2.1. N1 (140–160 ms)

There were four-way interaction effects across the hemisphere\emotion regulation\anticipated emotion\group, *F*(2,142) = 4.96, *p* = 0.01, η_p_^2^ = 0.65, and a significant difference between the groups, *F*(1,71) = 5.34, *p* = 0.024, η_p_^2^ = 0.07. A further simple effect analysis was conducted within the context of HAV, which revealed that the average amplitude elicited by the environmental message framing was more negative in the LAV circumstances (see Figure 3 and Figure 4).

The results of N1 showed that the average amplitude was more negative in the LAV environment. Specifically, as shown in Figure 4, in the HAV context, increasing and decreasing anticipated pride in the left and midline regions caused negative tendencies to increase and decrease the average amplitude of N1, while it was the opposite for guilt in the right brain region. The results of the anticipated guilt average amplitude—elicited by the increase and decrease in anticipated emotions by the environmental message framing in the HAV context—were the opposite of those observed for pride (see Table 4). In other words, by increasing and decreasing anticipated emotions through the environmental framing, the trend of the average volatility triggered by anticipated guilt and pride in the same aesthetic value environment, and that of the processing volatility of the same emotion in HAV and LAV environments, were opposite.

#### 3.2.2. P2 (200–280 ms)

There was no significant main effect within the subject’s effect for P2. The ANOVA result revealed the effects of two-way interactions between hemisphere and self-conscious emotion, *F*(2,142) = 6.62, *p* = 0.003, η_p_^2^ = 0.085. Moreover, there were three-way interaction effects between hemisphere, self-conscious emotion, and emotion regulation, *F*(2,142) = 4.22, *p* = 0.019, η_p_^2^ = 0.056. There was a significant effect of between-subjects variables for HAV and LAV.

A simple analysis indicated that there was no significant effect regarding HAV, in either increasing or decreasing anticipated pride or guilt, whereas regarding LAV, the average amplitude of increasing guilt in the right (3.57 ± 0.53 µV) was the largest compared to the left (2.9 ± 0.49 µV) and midline (2.97 ± 0.6 µV). The difference between the average amplitude between the midline and the right side was significant, *p* = 0.042; and decreasing guilt in the right (3.59 ± 0.57 µV) was the largest compared to the left (2.97 ± 0.53 µV) and midline (2.98 ± 0.64 µV). The difference between the midline and the right side was significant, *p* = 0.05 (see Table 3). The average amplitude of arousal regarding HAV was larger than that regarding LAV (see Figure 5 and Table 5). 

The results of P2 showed that the average amplitude for P2 was larger in the LAV group than in the HAV group.

#### 3.2.3. N2 (270–330 ms)

The main effect on the hemisphere revealed a significant effect within the subjects effect for P2, *F*(2,142) = 25.41, *p* < 0.001, η_p_^2^ = 0.26. The ANOVA results revealed that the two-way interaction effects between hemisphere and group (HAV vs. LAV) were significant, *F*(2,142) = 3.37, *p* = 0.049, η_p_^2^ = 0.045. There were interaction effects between hemisphere and emotion regulation (increasing vs. decreasing), *F*(2,142) = 9.85, *p* < 0.001, η_p_^2^ = 0.12. There were two-way interaction effects between hemisphere and anticipated emotion (AP vs. AG), *F*(2,142) = 13.46, *p* < 0.001, η_p_^2^ = 0.16. Importantly, there were three-way interaction effects among hemisphere, self-conscious emotion, and emotion regulation, *F*(2,142) = 7.33, *p* = 0.001, η_p_^2^ = 0.09, as well as among the hemisphere and self-conscious emotion, and group, *F*(2,142) = 3.25, *p* = 0.05, η_p_^2^ = 0.044.

There was a significant effect between the subject variables of HAV and LAV, *F*(1,71) = 11.43, *p* = 0.001, η_p_^2^ = 0.149 (see Figure 6).

The simple analysis further revealed significant differences regarding HAV for anticipated pride and guilt, whereas with increasing and decreasing the anticipated emotion, when participants received the message about increasing pride, they increased the average amplitude in the left area (3.17 ± 0.61 µV). The average amplitude was largest compared with the midline (2.04 ± 0.74 µV) and right side (2.72 ± 0.65 µV). The difference between the left-lateralized area and midline was significant, *p* < 0.001. Notably, when the participants received the message about decreasing anticipated pride, the average amplitude increased in the right-lateralized area (2.79 ± 0.62 µV), which was the largest compared to the midline (1.7 ± 0.75 µV) and left side (2.54 ± 0.64 µV). The difference in the average amplitudes between the right-lateralized area and midline was significant, *p* = 0.016.

When the participants received the message regarding the increase in emotion regulation for anticipated guilt, the average amplitude triggered on the right-lateralized area (3.14 ± 0.6 µV) was the largest compared to the midline (2.03 ± 0.7 µV) and left side (2.55 ± 0.6 µV). The difference in the average amplitudes between the right-lateralized area and midline was significant, *p* = 0.001. When the messages were decreased, the average amplitude increased on the right-lateralized area (3.15± 0.64 µV), which was the largest compared with the midline (2.15 ± 0.7 µV) and left side (2.5 ± 0.59 µV). The difference in the average amplitudes between the right-lateralized area and midline was significant, *p* = 0.001.

Regarding LAV, when the participants received the message of increased anticipated pride, the average amplitude on the right-lateralized area (0.18 ± 0.66 µV) was the largest compared to the left-lateralized area (0.002 ± 0.62 µV) and midline (−1.49 ± 0.75 µV) side. The differences in the average amplitudes between the midline and the right side, and between the midline and left were significant. Both values of difference were *p* < 0.001, and that between the left and the right side was not significant, *p* = 1. When the participants received the message of decreased anticipated pride, the average amplitude in the right-lateralized area (0.4 ± 0.63 µV) was the largest compared to the left-lateralized area (−0.68 ± 0.65 µV) and midline (−1.81 ± 0.76 µV) side, and the difference between the middle and the right side and that between the midline and left were significant. Both values were *p* < 0.001, and the difference between the left and the right side was significant, *p* = 0.025 (see Table 6).

For anticipated guilt, when the participants received the message of increased anticipated guilt, the triggered average amplitude on the right (0.69± 0.61 µV) was the largest compared to the left-lateralized area (−0.21 ± 0.61 µV) and midline (−1.36 ± 0.71 µV) side. The difference in the average amplitude between the midline and the right side, and that between the midline and left were significant, both were *p* < 0.001. The difference between the left and right sides was not significant, *p* = 0.082 (see Table 6). The average amplitude of increasing guilt in the right (1.11 ± 0.65 µV) was the largest compared to the left (0.27 ± 0.59 µV) and midline (−0.83 ± 0.71 µV) side. The differences between the midline and the right side and between the midline and left were significant (both values were *p* < 0.001), and the difference between the left and the right side was significant, *p* = 0.001. The average amplitude of arousal regarding HAV was larger than that for LAV (see Figure 6).

The average amplitude of ERPs for N2 revealed that, regarding HAV, the amplitude of N2 was larger than that in the LAV group, and there was a negative amplitude in the LAV group.

#### 3.2.4. LPP (400–800 ms)

There was no significant main effect in the ANOVA results. There were three significant two-way interaction effects, namely, interaction effects between hemisphere and group, *F*(2,142) = 3.61, *p* = 0.044, η_p_^2^ = 0.048, and between hemisphere and emotion regulation (increasing vs. decreasing), *F*(2,142) = 20.47, *p* < 0.001, η_p_^2^ = 0.224, as well as between hemisphere and anticipated emotion, *F*(2,142) = 40.93, *p*< 0.001, η_p_^2^ = 0.366. Importantly, there were significant three-way interaction effects among hemisphere, self-conscious emotion, and emotion regulation, *F*(2,142) = 46.08, *p* < 0.001, η_p_^2^ = 0.39. There were significant four-way interaction effects between hemisphere, self-conscious emotion, emotion regulation, and group (HAV vs. LAV), *F*(2,142) = 4.79, *p =* 0.015, η_p_^2^ = 0.063

There was a significant difference in LPP average amplitude between the HAV and LAV groups, *F*(1,71) = 22.37, *p* < 0.001, η_p_^2^ = 0.24 (see Figure 7).

Further simple analysis was conducted in the context of the HAV environment, and when the participants received the message of increasing anticipated pride, the average amplitude triggered by the emotion regulation was the largest in the left brain area (5.09 ± 4.34 µV), and the amplitude of the right side (3.01 ± 4.06 µV) was smallest. The difference in the average amplitudes between left and right was significant, *p* < 0.001. The volatility in the middle (3.85 ± 468 µV) was larger than that on the right. In the same manner, when participants received the message of increasing anticipated guilt, the largest amplitude was in the right-lateralized area (3.6 ± 4.33 µV), and the smallest was in the left-lateralized area (3.03 ± 4.34 µV). The amplitude in the midline (3.26 ± 4.98 µV) was larger than that in the left-lateralized area; there was no significant difference.

Regarding HAV, when the participants received the message of decreased anticipated pride, there was no significant difference in the average amplitudes triggered by the emotion regulation among the hemispheres (see Table 7). In the HAV context, the difference in the lateralization of the left-middle-right brain regions was significant only when anticipated pride was increased. In other emotion regulation situations (increasing anticipated guilt, decreasing anticipated pride, and decreasing anticipated guilt), the difference in lateralization processing was not significant.

Regarding LAV, when the participants received the message of decreasing anticipated pride, the average amplitude triggered by the emotion regulation in the midline area (−0.71 ± 3.36 µV) was larger than in either the left-lateralized (0.1 ± 2.99 µV) or right-lateralized (−0.4 ± 3.02 µV) area. The difference in the average amplitudes between the midline and left was significant (*p* = 0.016), and the difference between the left and right sides was not significant. When the anticipants received the message of decreasing anticipated pride, the average amplitude in the midline area (−1.13 ± 3.73 µV) was the largest compared to the left-lateralized area (−0.81 ± 2.87 µV) and right-lateralized area (0.05 ± 3.4 µV). The difference between the midline and the right side was significant, *p* < 0.001.

When increasing anticipated guilt, the average amplitude triggered in the midline area (−0.7 ± 3.71 µV) was larger than in either the left-lateralized (−0.57 ± 3.02 µV) or right-lateralized (0.41 ± 3.28 µV) area. The difference in the average amplitudes between the middle and the right side was significant, *p* < 0.001, and the difference between the left and the right side was significant, *p* = 0.031.

When participants were exposed to the HAV environment, they received the environmental message framing of emotion regulation (increasing or decreasing the anticipated pride), which triggered the average amplitude of LPP to change direction (increasing and decreasing the average amplitude), consistent with the direction of emotion regulation in midline and right areas (see Figure 8). Regarding the LAV environment, the changing direction was the opposite.

The findings suggest that ERPs of emotion regulation processing were lateralized to the left and right hemispheres (see Table 7 and Figure 7). Specifically, when the participants received the environmental message framing of increasing the anticipated pride, the largest amplitude triggered was generally left-lateralized. When they received the message of decreasing anticipated guilt, it was generally left-lateralized. However, there was a significant difference between the emotion-regulation of self-consciousness for the anticipated pride and guilt in the temporal dynamics of decision-making information processing.

The ERP results showed that in the HAV context, increasing and decreasing anticipated pride corresponded to an increase and a decrease in LPP amplitudes in the left and middle regions, respectively, while increasing and decreasing anticipated pride in the right hemisphere showed the opposite trend. While the trends of increasing and decreasing anticipated guilt in the left- and middle-brain regions were opposite, only the right side increased. Moreover, decreased anticipated guilt increased and decreased the LPP average amplitude (see Table 7).

The results demonstrated that the emotion regulation adjusted the average amplitude of the LPP and altered their realistic pro-environment behavior.

## 4. Discussion

Our study explored the effects of anticipated emotion regulation and aesthetic value environment on pro-environmental behavior and its underlying neural mechanisms. The behavioral results, which found that people’s pro-environmental behaviors can be regulated by regulating anticipated emotions, validate the effect of self-conscious emotions on pro-environmental behaviors [11,48,60]. Importantly, they revealed that people in the HAV or LAV environment could regulate anticipated emotions by the environmental message framing. When they regulated self-conscious emotions, such as anticipated pride and guilt, they would regulate individuals’ pro-environmental behaviors.

Specifically, the behavior results are consistent with those of previous studies on pro-environmental intention, which found that increased pride and decreased guilt by positive framing could increase purchase intention for green products, whereas increased guilt and decreased pride can reduce it through negative framing [55]. The results of pro-environmental behavior verify that people’s pro-environmental behavior could be influenced by the regulation of anticipated emotions from the perspective of individual-specific pro-environmental behaviors, and they expand the research from pro-environmental intention to behavior.

The findings also extend research on emotion regulation. Scholars have reported that self-conscious emotions have a self-regulating function and that emotion regulation would be applied to the emotions being experienced [25,56] and to the regulation of anticipated emotions, and its effectiveness can be reflected in participants’ changes in specific behavioral tendencies and differences in ERP components. Our research confirms that anticipated emotions can influence people’s decision-making [30].

In contrast to previous studies, these findings revealed that the internal factors of anticipated emotion influences behavior by activating temporal processing in different brain regions through ERP measures. It suggests that the aesthetic value environment can influence emotion regulation by increasing and decreasing the amplitude of the central or posterior lateralized area from different hemispheres by environmental message framing. This internal processing was manifested as a change in pro-environmental behavior in explicit behavior. It was consistent with the previous study of changes in LPP amplitudes in emotion regulation of the LPP [61]. The results expand the specific time course and regularity of the lateralized processing of anticipated emotion regulation; that is, the largest average amplitude of increasing anticipated pride was located in the left posterior brain area, which had the same average amplitude when participants received the message of decreasing anticipated guilt. It suggests that the aesthetic value of environmental cues perceived by people could affect future anticipated emotional processing, which manifests mainly from the differences in each component of ERP processing located in the central and posterior brain regions from 140 ms to 800 ms. Specifically, it was embedded in the biased processing of brain regions. The experimental results showed that increasing and decreasing anticipated emotions will affect the change in excitation amplitude. Most importantly, it demonstrates that people’s perceptions of the aesthetic value of the external physical environment can have an intrinsic influence on anticipated emotional experiences, as manifested in explicit behaviors.

The results also indicated that the aesthetic value of the physical environment situation affects people’s pro-environmental behavior. However, the difference in the pro-environmental behavior between the HAV and LAV environments was not significant. The findings were not consistent with those of previous studies. The reason could be the role of emotion regulation, as we have conducted emotion regulation in HAV and LAV environments and have found changes in pro-environmental behavior in the emotion regulation process. The emotion regulation interacted with the perceptions of the HAV and LAV environments, which together regulated an individual’s pro-environmental behavior [55].

## 5. Conclusions

This study examined the effects of increasing and decreasing anticipated emotions through environmental framing to anticipate the future impact on individuals’ participation in pro-environmental behaviors across the varying contexts of HAV and LAV. The behavioral results verified the hypothesis that increased anticipated emotions among individuals can increase their pro-environmental behavior and decrease anticipated emotions, thus reducing individuals’ pro-environmental behaviors. These behavioral results reaffirm the effects of anticipated guilt and pride on pro-environmental behaviors.

The results show that in the HAV situation, the processing of anticipated pride and guilt is significantly different in the left brain area, while in the low-aesthetic-value situation, the participants’ processing of anticipated pride and guilt is significantly different in the right brain area. Moreover, the difference between increasing pride and guilt in HAV situations is significant, while in LAV situations, there is a significant difference between reduced anticipated pride and guilt. From this result, we draw the following conclusions: individuals have significant differences in the anticipated emotion regulation processing in HAV and LAV situations, and their internal processing mechanisms are different.

These findings have practical significance in promoting environmental protection activities in daily life. Thus, corresponding environmental pictures can be set across different life situations and align with the appropriate environmental framing texts to ensure the improved promotion of pro-environmental behaviors in peoples’ daily lives as practical operations are feasible and easy to implement.

This study considered only the interaction effects of the HAV and LAV environments and emotion regulation on pro-environmental behavior, and did not explore the influence of the two variables of aesthetic value environment and emotion regulation on pro-environmental behavior, respectively. Future studies should explore the changing footprint of pro-environmental behavior from the perspectives of aesthetic value environment and emotion regulation, and how these variables affect pro-environmental behavior.

## Figures and Tables

**Figure 1 ijerph-19-05714-f001:**
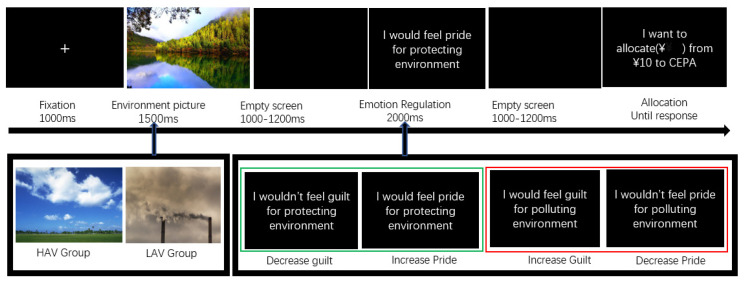
Trail structure of the experiment. Content in green boxes indicate that positive environmental framing increases anticipated pride and reduces anticipated guilt, while red boxes indicate that negative environmental framing increases anticipated guilt and reduces anticipated pride.

**Figure 2 ijerph-19-05714-f002:**
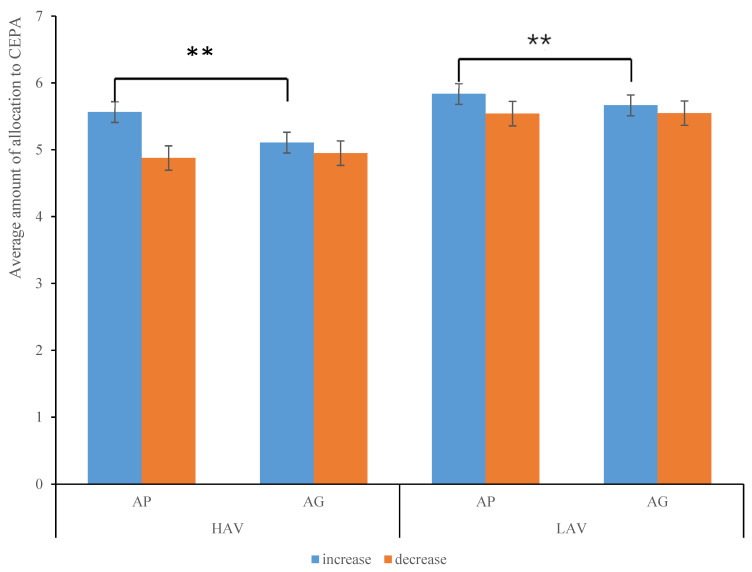
The descriptive statistics bar graph of the average allocation amount of aesthetic value environment. ** *p* < 0.01.

**Figure 3 ijerph-19-05714-f003:**
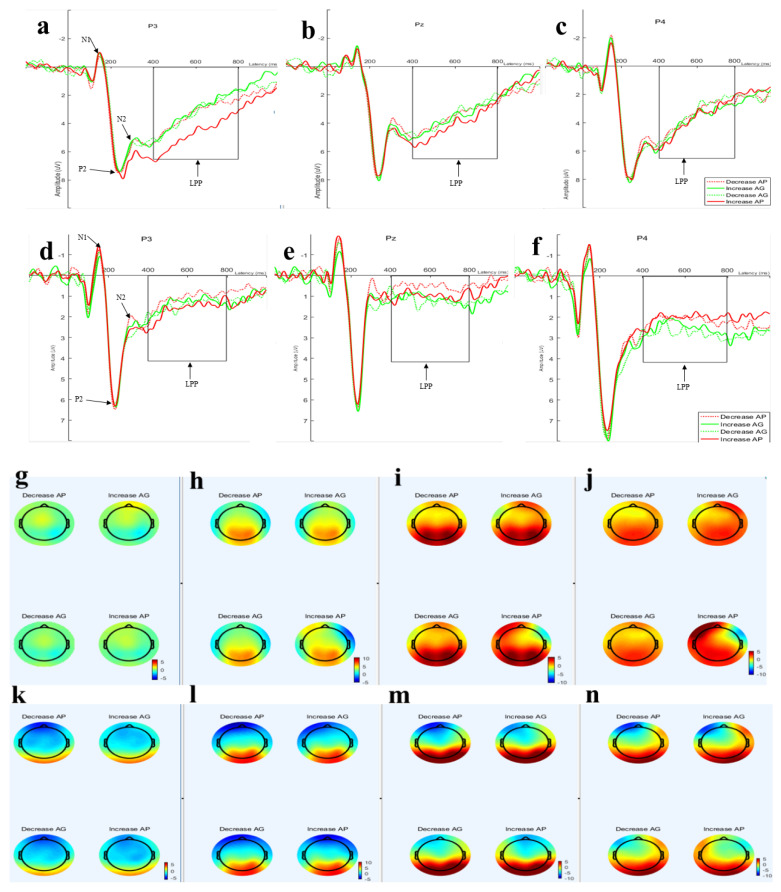
ERP components to emotion regulation at central-posterior. (**a**–**c**) Waveforms representing the average amplitude of the electrode points in the left (**a**: P3), midline (**b**: PZ), and right (**c**: P4) brain regions, respectively, under the context of the HAV; and (**d**–**f**) the LAV, the electrode points in the left (**a**: P3), midline (**b**: PZ), and right (**c**: P4) brain regions, respectively. The voltage topography (**g**–**j**) is displayed in the corresponding time windows, showing the scalp distribution at 140–160 ms (**g**: N1), 200-280 ms (**h**: P2), 270-330 ms (**i**: N2), and 400-800 ms (**j**: LPP) within the context of the HAV, and (**k**: N1)\(**l**: p2)\(**m**:N2)\(**n**: LPP) within the context of LAV.

**Figure 4 ijerph-19-05714-f004:**
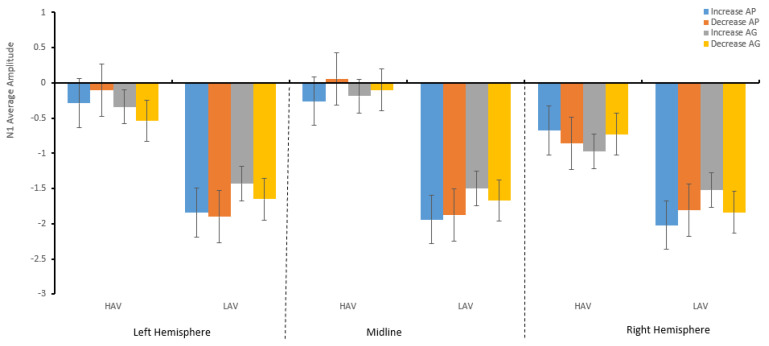
The descriptive statistics bar graph of the average amplitude of N1 regarding the hemisphere and emotion regulation.

**Figure 5 ijerph-19-05714-f005:**
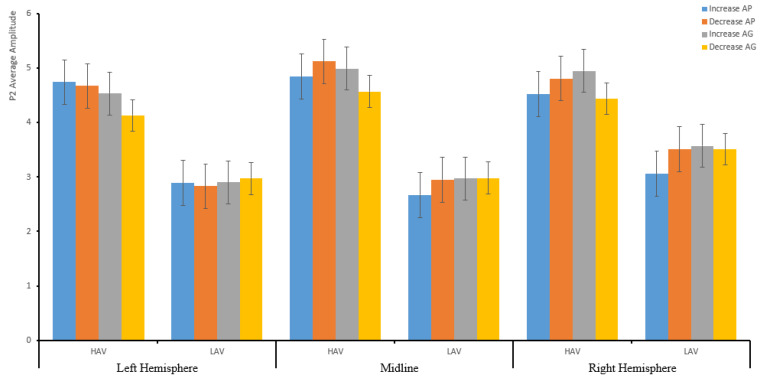
The descriptive statistics bar graph of the average amplitude of P2 about the hemisphere and emotion regulation.

**Figure 6 ijerph-19-05714-f006:**
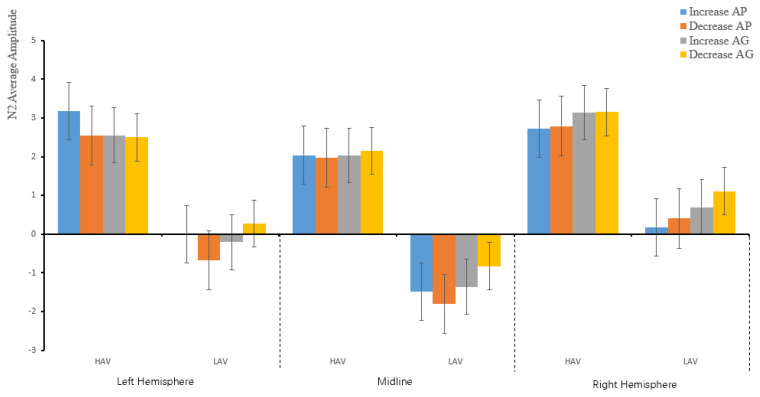
The descriptive statistics bar graph of the average amplitude of N2 about the hemisphere and emotion regulation.

**Figure 7 ijerph-19-05714-f007:**
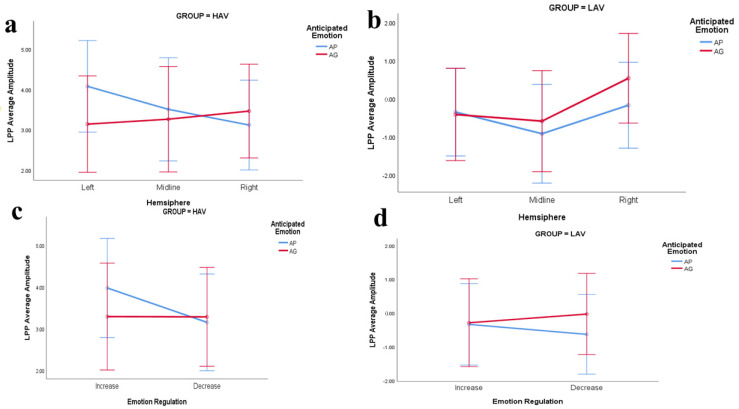
The interaction effects of the hemisphere, emotion regulation, and anticipated emotion for LPP. (**a**,**c**) The interaction of the hemisphere, emotion regulation, and anticipated emotion for LPP under the context of the HAV. (**b**,**d**) The interaction under the context of the LAV.

**Figure 8 ijerph-19-05714-f008:**
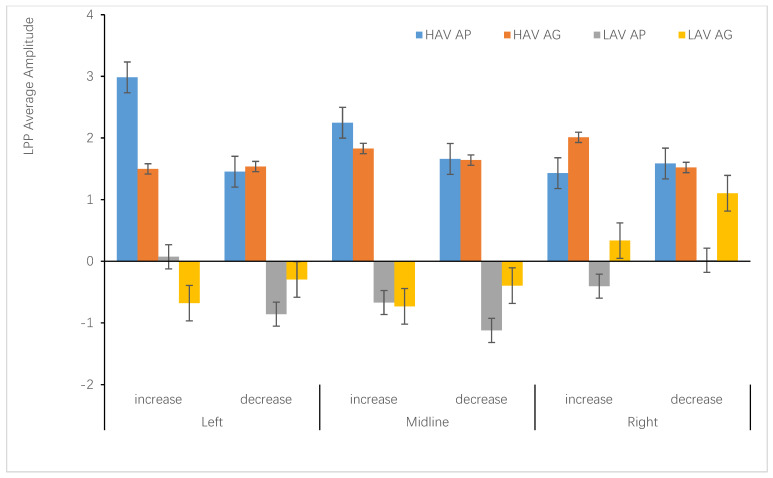
The descriptive statistics bar graph of the average amplitude of LPP regarding the hemisphere and emotion regulation.

**Table 1 ijerph-19-05714-t001:** Aesthetic rating criteria.

Criteria	Rating Prompt	Low Anchor	High Anchor
Complex	This space looks…	Simple	Complex
Order	This space looks…	Disordered	Organized
Beauty	This space looks…	Ugly	Beautiful
Comfort	This makes me feel…	Uncomfortable	Comfortable
Valence	This makes me feel…	Bad	Good
Relaxation	This makes me feel…	Stressed	Relaxed

**Table 2 ijerph-19-05714-t002:** Material evaluation of the aesthetic value environment.

	HAV		LAV		*t*	*df*	*p*	*d*
	*M*	*SD*	*M*	*SD*				
Complex	3.82	1.34	4.06	1.46	−0.91	48	0.37	−0.13
Order	5.79	0.87	2.16	0.62	21.94	48	≤0.001	3.13
Beauty	5.95	0.86	1.88	0.59	24.51	48	≤0.001	3.5
Comfort	5.99	0.84	1.87	0.61	24.92	48	≤0.001	3.55
Valence	6.02	0.86	1.91	0.63	24.06	48	≤0.001	3.43
Relaxation	6.05	0.82	2.14	0.72	23.15	48	≤0.001	3.31

**Table 3 ijerph-19-05714-t003:** Allocation amounts (mean ± SD): Increase AP signifies when participants received the message of increasing anticipated pride (“I would feel pride in protecting the environment”), and Decrease AP signifies when participants received a message of decreasing anticipated pride (“I wouldn’t feel pride in polluting the environment”). Similarly, for Increase AG, the message was “I would feel guilty about polluting the environment”, and for Decrease AG, the message was “I wouldn’t feel guilty about protecting the environment”.

Aesthetic Value	Increase AP		Decrease AP		Increase AG		Decrease AG	
	*M*	*SD*	*M*	*SD*	*M*	*SD*	*M*	*SD*
HAV	5.57	2.24	4.88	2.13	5.11	2.14	4.95	2.15
LAV	5.84	2.56	5.54	2.6	5.67	2.58	5.55	2.62

**Table 4 ijerph-19-05714-t004:** N1 amplitudes (mean and standard error).

Hemisphere	Aesthetic Value	Increase AP		Decrease AP		Increase AG		Decrease AG	
		*M*	*SD*	*M*	*SD*	*M*	*SD*	*M*	*SD*
Left	HAV	−0.29	2	−0.1	2.42	−0.34	2.43	−0.54	2.15
	LAV	−1.84	2.44	−1.9	2.64	−1.43	2.25	−1.65	2.08
Midline	HAV	−0.26	2.87	0.06	3.34	−0.19	3.17	−0.1	3.34
	LAV	−1.94	3.55	−1.88	3.91	−1.5	3.23	−1.67	3.33
Right	HAV	−0.68	2.57	−0.86	2.55	−0.97	2.39	−0.73	2.67
	LAV	−2.02	3.33	−1.81	3.33	−1.52	2.98	−1.84	3.21

**Table 5 ijerph-19-05714-t005:** P2 amplitudes (mean and standard error).

Hemisphere	Aesthetic Value	Increase AP		Decrease AP		Increase AG		Decrease AG	
		*M*	*SE*	*M*	*SE*	*M*	*SE*	*M*	*SE*
Left	HAV	4.74	0.52	4.67	0.55	4.53	0.48	4.13	0.52
	LAV	2.89	0.52	2.83	0.56	2.9	0.49	2.97	0.53
Midline	HAV	4.84	0.63	5.12	0.66	4.99	0.59	4.57	0.63
	LAV	2.67	0.64	2.95	0.66	2.97	0.6	2.98	0.64
Right	HAV	4.52	0.56	4.81	0.56	4.95	0.52	4.44	0.56
	LAV	3.06	0.57	3.51	0.57	3.57	0.53	3.59	0.57

**Table 6 ijerph-19-05714-t006:** N2 amplitudes (mean and standard error).

Hemisphere	Aesthetic Value	Increase AP		Decrease AP		Increase AG		Decrease AG	
		*M*	*SE*	*M*	*SE*	*M*	*SE*	*M*	*SE*
Left	HAV	3.17	0.61	2.54	0.64	2.55	0.6	2.5	0.59
	LAV	0.002	0.62	−0.68	0.65	−0.21	0.61	0.27	0.59
Midline	HAV	2.04	0.74	1.97	0.75	2.03	0.7	2.15	0.7
	LAV	−1.49	0.75	−1.81	0.76	−1.36	0.71	−0.83	0.71
Right	HAV	2.72	0.65	2.79	0.62	3.14	0.6	3.15	0.64
	LAV	0.18	0.66	0.4	0.63	0.69	0.61	1.11	0.65

**Table 7 ijerph-19-05714-t007:** Late positive potential (LPP) amplitudes (mean and standard error).

Hemisphere	Aesthetic Value	Increase AP			Decrease AP			Increase AG			Decrease AG		
		*M*	*SD*	*SE*	*M*	*SD*	*SE*	*M*	*SD*	*SE*	*M*	*SD*	*SE*
Left	HAV	5.09	4.34	0.62	3.07	4.07	0.58	3.03	4.74	0.66	3.25	4.3	0.61
	LAV	0.1	2.99	0.62	−0.81	2.87	0.59	−0.57	3.03	0.67	−0.26	2.94	0.62
Midline	HAV	3.85	4.68	0.67	3.17	4.36	0.67	3.26	4.98	0.72	3.27	4.22	0.67
	LAV	−0.71	3.36	0.68	−1.13	3.73	0.68	−0.7	3.71	0.68	−0.48	3.85	0.67
Right	HAV	3.01	4.06	0.59	3.23	3.66	0.58	3.6	4.33	0.63	3.34	3.77	0.6
	LAV	−0.4	3.02	0.6	0.05	3.4	0.59	0.41	3.28	0.64	0.66	3.46	0.6

## Data Availability

The data presented in this study are available on request from the corresponding author. The data are not publicly available, due to concerns about privacy and ethics in personal decision-making.

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
