# Peer review of "Influence of Environmental Aesthetic Value and Anticipated Emotion on Pro-Environmental Behavior: An ERP Study"

_ijerph, 2022, doi:10.3390/ijerph19095714_

Round 1

Reviewer 1 Report

Thank you for the opportunity to review your manuscript.  At present, the write-up requires considerable revision, both for clarification and correction. In reporting of results, it seems that you are claiming differences or effects where there were no significant results. I commend the efforts to report these complex findings, but additional work is needed to get them into an acceptable and readily comprehensible form.  I have combined my marked up copy of the file together with notes. 

Author Response

28 April 2022

Dear expert,

We are writing in reference to our International Journal of Environmental Research and Public Health (IJERPH)submission: “The Influence of Environmental Aesthetic Value and Anticipated Emotion on Pro-environmental Behavior: An ERP Study” [ijerph-1679579].

First of all, we thank you and the two reviewers very much for the recognition of the merits of our revision. We further appreciate your’ valuable feedback and for giving us one more precious opportunity to revise the manuscript. We have dedicated our best efforts to address the comments in this resubmission.

Below, we reproduce the reviewer’s comments, with our responses and changes in the revised manuscript in blue font. Please let us know if there is anything else we could do to improve the manuscript.

Best wishes,

Yiping Zhong,

Professor of Department of Psychology,

Hunan Normal University, Changsha, P.R. China.

Reviewer 1:

IJERPH REVIEW ijerph-1679579

13 April 2022 Article. 

The Influence of Environmental Aesthetic Value and Anticipated Emotion on Pro-environmental Behavior: An ERP Study

Thank you for the opportunity to review your manuscript.  At present, the write-up requires considerable revision, both for clarification and correction. In reporting of results, it seems that you are claiming differences or effects where there were no significant results. I commend the efforts to report these complex findings, but additional work is needed to get them into an acceptable and readily comprehensible form.  

I have some specific comments either because there are issues that must be addressed or for you to take into consideration. 

Comments and response:

Abstract 

lines 17-18. pro-environmental, and seems to be a word missing: “by integrating behavioral and using” 

Response: Thank you very much for pointing this out. We have revised the term used here.

“This study investigated the neural mechanisms underlying interaction effects between environmental context and emotion regulation on pro-environmental behavior by integrating behavioral and temporal dynamics of decision-making information processing with the event-related potential (ERP) technique measures.”

Introduction

Lines 34-36. Seems the order should be the other way around. i.e., First, the environment needs to be protected well…” Second, in order to protect the environment…” 

Response: Thank you very much for your suggestion.

We rewrote the sentence as follows: First, regarding a high aesthetic value (HAV) environment, to ensure that the environment continues to be suitable for human survival, actions towards its protection should be prioritized. Second, the environment needs to be protected well because human beings cannot survive in polluted and destroyed environments.

Line 41. “little to no harm or benefits” can be taken to read little to no harm and little to no benefit. I suggest rewording for clarity; e.g., “refers to behavior that either induces little to no harm or that benefits the environment” 

Response: Thank you very much for your pertinent suggestion.

We have revised the line according to your suggestion: It refers to behavior that induces little to no harm or that benefits the environment.

Lines 45 & 47. engagement in, not with

Response: Thank you very much for your suggestion.

We have revised the line according to your suggestion: According to the perception-emotion-behavior model, the reasons for human engagement in pro-environmental behaviors may be motivated by external environmental cues or by internal motivations.

Lines 51-52. suggest reword for clarity; e.g., “participants with a stronger intention to engage in moral behavior…”

Response: We thank you for your suggestion.

We have revised the line according to your suggestion: Studies have suggested that participants with a stronger intention of engaging in moral behavior in an environment were associated with having a HAV, whereas a decreased intention was associated with a low aesthetic value (LAV).

Lines 64-67. Needs rewording for clarity. “Meaning the emotions…” is not a complete sentence. 

Response: We thank you for suggestion.

 We have revised the line according to your suggestion: Hence, the emotions people experience while in the physical environment, will be similar to the imagined future environment.

Line 67. What do you mean by “emotions that have not emerged”? In the subconscious? If they haven’t ‘emerged’ (in what sense?), how can they have an impact? 

Response: We thank you for your comment.

We apologize for the unclear statement. We have revised the line as follows:

The emotions people have not experienced would have a significant impact on the individual’s cognitive processes [28], behaviors [29], and decision-making processes [30], as same as the experienced emotion.

Lines 69-72. Split into two sentences for clarity. 

Response: We thank you for your comment and suggestion.

We revised the sentence into two sentences as follows: Anticipated feelings, such as pride and guilt, are often regarded as self-conscious emotions. The self-conscious emotion is a special type of emotion that involves the ego, and it is particularly relevant to comprehending environmentally responsible decision-making processes or behaviors consistent with the Norms Activation Model (NAM).

Line 82. The “Therefore” here doesn’t follow. The previous sentence refers to emotion stemming from past or failed action; this is not anticipated but rather is retrospective. 

Response: We thank you for your comment and suggestion.

We have deleted the inappropriate word.

Lines 84-85. “have described” doesn’t fit with “is known”, and should be “based on”, not from Line 86. Do you mean ‘anticipatory thinking’? 

Response: We thank you for your comment and suggestion.

We revised the section according to your suggestion as follows: Scholars have described anticipated emotional response based on the prospect of success and failure as pre-evaluation, also known as pre-thinking or anticipated thinking.

Lines 89-90. Well, there are many potential influencing factors for different behaviours and/or inconsistencies in behaviours, including societal structures that either support or hinder PEB. 

Response: We thank you for your comment and suggestion.

We added  references for the factors potentially influencing different behaviors and/or inconsistencies in behaviors, as follows: Moreover, research has suggested that different types of information could either promote or dampen pro-environmental behavior and subsequent spillover effects [1]. Another finding regarding the strong support for environmental protection was contrasted by low actual pro-environmental behavior [2].

Line 104. “which was proved by different ERP components” needs rewording. Could be “supported by” 

Response: We thank you for your comment and suggestion.

We revised the sentence, according your suggestion, as follows: . . . which was supported by different ERP components, N1\P2\late positive potential (LPP) in the three stages of time processing.

Line 114. “pictures of positive emotions” needs explanation. Do you mean, for instance, images of facial expressions portraying positive emotions? 

Response: We thank you for your comment and suggestion.

We apologize for our unclear statement. We have revised it as follows: Evidence revealed that environmental pictures of positive emotions presented before an individual’s decision were sufficient to neutralize the framing effect by reducing the aversion aspect of affirmative loss.

Line 119. “Whereas the maintain-negative condition…” is not a sentence. 

Response: We thank you for pointing this out.

We have revised the sentence as follows: . . . while maintaining the negative emotion condition elicited a more minimal LPP amplitude.

Line 130. “positive and negative emotional backgrounds” means what? Needs clear explanation. 

Response: We thank you for your comment and suggestion.

We rewrote the hypothesis as follows: Two hypotheses are tested in this study: First, there is a difference between the environmental behavior of individuals in HAV and LAV environments, with the LAV environment expected to present more pro-environmental behavior. Second, in the process of increasing and decreasing anticipated emotions through environmental framing, neural processes are associated with individuals' environmental behaviors, and the ERP components are changed by different tendencies under the circumstance of the HAV and LAV

Line 136. Where are environmental aesthetic values addressed in hypotheses? 

Response: We thank you for your comment and suggestion.

We rewrote the hypotheses as follows: Two hypotheses are tested in this study: First, there is a difference between the environmental behavior of individuals in HAV and LAV environments, with the LAV environment expected to present more pro-environmental behavior. Second, in the process of increasing and decreasing anticipated emotions through environmental framing, neural processes are associated with individuals' environmental behaviors, and the ERP components are changed by different tendencies under the circumstance of the HAV and LAV.

Line 137. You need to explain how you operationalized your DV. The DV appears to have been the amount of cash allocated to the environmental organization. Clarifying this here would help to avoid later confusion when the environmental organization is first mentioned. 

Response: We thank you for your comment and suggestion.

We apologize for the lack of clarity in our explanation regarding how we operationalized our DV. We have revised the methodology as follows: In this study, pro-environmental behavior refers to the specific amount that participants allocate to environmental organizations during the course of the experiment in a monetary allocation task related to their own reward for participating in the experiment.

Line 142. Are you suggesting aesthetic values as a proxy for ‘emotional background’? This needs clarification/explanation. 

Response: We thank you for your suggestion.

We have revised the text as follows: Among the participants, 37 were assigned to the HAV group, who initiated environmental situations by viewing HAV environmental pictures and 36 to the LAV group to view the LAV environmental pictures.

Lines 151-152. “natural environmental pictures that were well-taken kept” – “Half of the pictures portrayed well-preserved natural environments”. “displayed environmental pollution” – “The remaining 32 pictures portrayed polluted environments” 

Response: We thank you for your comment and suggestion.

We have revised the sentence as follows: Half of them portrayed well-preserved natural environments, and the remaining 32 portrayed polluted environments (including air and water pollution) with obvious traces of human activities, without environmental protection.

Line 164. Delete ‘showed’ 

Response: We thank you for your suggestion.

We deleted ‘showed’ in the concerned text.

Table 2. Table content alignment needs tidying up. 

Response: We thank you for suggestion, and have corrected our table alignment accordingly

Lines 177-179. Needs clarification. Do you mean that some environmental organizations provided project funding? Need to explain that this feature served as your PEB DV. 

Response: We thank you for your comment and suggestion.

We have revised the section as follows: When introducing the experimental process, we first presented the scientific research projects between the environmental protection organizations and our laboratory. Our study was supported by this research project.  The research task was to examine people's sensitivity to money. The specific task was to allocate money tasks; each distribution would be related to the participants’ rewards and public welfare funds raised by environmental protection organizations. The specific tasks were that the participant would get 10 RMB in each round of assignment, and they could assign any amount (0-10) to the environmental protection organization, and keep the rest.

Lines 182-183. Avoid gender bias (i.e., he, himself) in writing; can use gender neutral ‘they’ or ‘themselves’. 

Response: We thank you for your pertinent suggestion.

We revised the sentence as follows: The specific tasks were that the participant would get 10 RMB in each round of assignment, and they could assign any amount (0-10) to the environmental protection organization, and keep the rest.

Line 184. Past tense applies (We will inform). 

Response: We thank you for your comment and suggestion.

We rewrote the sentence as follows: To ensure the ecological validity of the experiment, we informed the participants: we should randomly select any number of allocation trials after completion of the task as their rewards.

Lines 184-185. This needs revision for clarity. It is not clear. 

Response: We thank you for your suggestion.

We rewrote the sentence as follows: To ensure the ecological validity of the experiment, we informed the participants: we should randomly select any number of allocation trials after completion of the task as their rewards.

Line 197. “they entered the amount...until they made their response.” This indicates they kept entering one or more cash amounts until a response was made - but what type of response and to what? Was the amount entered itself the response? In that case, you could indicate that they entered the amount they wanted to allocate and hit the 'enter' key to capture their response. This needs to be written more clearly.

Response: We thank you for your comment and suggestion. We have revised the sentence as follows: Finally, they input the amount of money that they want to allocate to the China Environmental Protection Association (CEPA) from the 10 yuan every trial. They could enter the next trial after they made their response (press key to enter numbers).

Line 204. So you indicated to participants that they had a hypothetical amount of 10 RMB for each of 240 trials (i.e. 2400 RMB total) from which to allocate, and they get the remainder (i.e. non-allocated amount) as a reward? So there is a disincentive to ever allocate the full amount, and also a disincentive to allocate nothing (social desirability effect, not wanting to look 'bad' to the investigators).  On average, what was the value of the reward accumulated by participants? 

Response: We thank you for your comment and suggestion.

We apologize for the lack of clarity, and have added the sentence: “On average, each participant was rewarded with 50 to 70yuan RMB”.

Figure 1. Have you considered the possibility that the positive images might elicit less PEB intention/behaviour simply because they don't indicate that any behaviour is required to 'help' that environment? It's akin to walking into a very clean/tidy room in a house; people would not be motivated to clean/tidy the room relative to one that was dirty/messy.

Response: We thank you for pointing this out.

We strongly agree with your suggestion, and after consulting relevant literature, we found references pertaining to your suggestions, which we have included in the Discussion.

Figure 1 re messages. The messages themselves differ in whether they refer to anticipated or current emotion. "I feel" indicates current emotion; "I wouldn't feel" indicates anticipated emotion.

To have the positive/increased effects reflect anticipated emotion, should have been "I would feel". 

Response: We thank you for your comment and suggestion.

We strongly agree with your suggestion, and have modified Figure 1 accordingly.

Lines 252-259. The write-up needs revision to align with the null findings. For example, there was no significant difference so it's incorrect to claim that one group engaged in more/less PEB than another. Must accept the null hypothesis. 

Response: We thank you for your comment.

We have deleted the null findings in the revised manuscript.

Figure 2. Need to indicate what the asterisks represent in the figure; it looks like you're indicating significant effects in places where no significant effects were found.  Also, what do the error bars represent (e.g., SE?).

Response: We thank you for your pertinent comment.

We have revised the Figure according to your suggestion.

Line 271. Refers to Figure 1. Do you mean Figure 2? But that's allocation amount. Seems you need another figure for RT results.

Response: We thank you for your comment and suggestion.

We apologize for our mistake concerning the Figure number. We have reordered the figures and carefully checked the order in which they are referenced in the text.

Line 295. Reference to Figure 3. Given the complexity of the figure, with 20 subcomponents, need to specify which parts of the figure refer. 

Response: We thank you for your comment and suggestion. We added relevant information to the Figure 3 legend as follows:

ERPs components to emotion regulation at central-posterior. (a)\(b)\(c) are the waveforms representing the average amplitude of the electrode points in the left (a: P3), midline (b: PZ) and right (c: P4) brain regions respectively under the context of the HAV; and the (d)(e)(f) are of the LAV, the electrode points in the left (a:P3), midline (b: PZ) and right (c:P4) brain regions respectively. The voltage topography (g)\(h)\(i)\(j) are displayed in the corresponding time windows of show the scalp distribution at the 140–160ms (g: N1), 200–280ms (h: P2), 270–330ms (i:N2), 400–800ms (j: LPP) within the context of the HAV, and the (k:N1) \ (l:p2) \ (m:N2) \ (n: LPP) within the context of LAV.

Section 3.2.3. This entire subsection needs revision for clarity and to correct errors (e.g., Figure 4 referred to, but should be Figure 5).

Response: We thank you for your comment and suggestion.

We apologize for our mistake about the Figure number. We have reordered the figures and carefully checked and corrected the order in which they are referenced in the text. We revised this entire subsection as follows:

The main effect on the hemisphere revealed a significant effect within the subjects effect for P2, F(2,142) = 25.41, p<0.001, ƞp2 = 0.26. The ANOVA results revealed that the two-way interaction effects between hemisphere and group (HAV vs LAV), F(2,142) = 3.37, p =0.049, ƞp2 = 0.045. There were interaction effects between hemisphere and emotion regulation (increasing vs decreasing), F(2,142) =9.85, p<0.001, ƞp2 = 0.12. There were two-way interaction effects between hemisphere and anticipated emotion (AP vs AG), F(2,142) = 13.46, p<0.001, ƞp2 = 0.16. Importantly, there were three-way interaction effects among hemisphere and self-conscious emotion, and emotion regulation, F(2,142) = 7.33, p =0.001, ƞp2 = 0.09 as well as among the hemisphere and self-conscious emotion, and group, F(2,142) = 3.25, p =0.05, ƞp2 = 0.044.

There was a significant effect between the subject variables of HAV and LAV, F(1,71) = 11.43, p =0.001, Æžp2 = 0.149 (see Figure 6).

The simple analysis further revealed significant differences regarding HAV for anticipated pride and guilt. Whereas increasing and decreasing the anticipated emotion, specifically, pride, increased the average amplitude on the left (3.17 ± 0.61 µV). The average amplitude was the largest compared with the midline (2.04 ± 0.74 µV) and right side (2.72 ± 0.65 µV). The difference between the left and midline was significant, p<0.001. Notably, when the participants received the message about decreasing anticipated pride, the average amplitude increased in right (2.79 ± 0.62 µV), which was the largest compared to the midline (1.7 ± 0.75 µV) and left side (2.54 ± 0.64 µV). The difference in the average amplitudes between the right and midline was significant, p=0.016.

When the participants received the message regarding the increase in emotion regulation for anticipated guilt, the average amplitude triggered on the right (3.14 ± 0.6 µV), was the largest compared to the midline (2.03 ± 0.7 µV) and left side (2.55 ± 0.6 µV). The difference of the average amplitudes between the right and midline was significant, p=0.001. When the messages were decreased, the average amplitude increased on the right (3.15± 0.64 µV), which was the largest compared with the midline (2.15 ± 0.7 µV) and left side (2.5 ± 0.59 µV). The difference in the average amplitudes between the right and midline was significant, p=0.001.

Regarding LAV, when the participants received the message of increased anticipated pride, the average amplitude on the right (0.18 ± 0.66 µV) was the largest compared to the left (0.002 ± 0.62 µV) and midline (-1.49 ± 0.75 µV) side. The differences in the average amplitudes between the midline and the right side, and between midline and left were significant. Both values of difference were p<0.001, and between the left and the right side was not significant, p =1. When the participants received the message of decreased anticipated pride, the average amplitude in the right (0.4 ± 0.63 µV) was the largest compared to the left (-0.68 ± 0.65 µV) and midline (-1.81 ± 0.76 µV) side, and the difference between the middle and the right side, and that between midline and left were significant. Both values were p<0.001, and the difference between the left and the right side was significant, p = 0.025 (see Table 6).

For anticipated guilt, when the participants received the message of increased anticipated guilt, the triggered average amplitude on the right (0.69± 0.61 µV) was the largest compared left (-0.21 ± 0.61µV) and midline (-1.36 ± 0.71 µV) side. The difference in the average amplitude between the midline and the right side, and that between the midline and left were significant, both were p < 0.001. The difference between the left and right sides was not significant, p = 0.082 (see Table 6). The average amplitude of increasing guilt in the right (1.11 ± 0.65 µV) was the largest compared to the left (0.27 ± 0.59 µV) and midline (-0.83 ± 0.71 µV) side. The differences between the midline and the right side, and between midline and left were significant, both values were p<0.001, and the difference between the left and the right side was significant p = 0.001. The average amplitude of arousal regarding HAV was larger than that for LAV (see Figure 6).

Line 386. Figure 6. There is no reference to Figure 6 within text; need to refer to this as relevant. 

Response: We thank you for your comment and suggestion.

We have reordered the figures and carefully checked and corrected the order in which they are referenced in the text.

Line 388. Table 6. There is no reference to Table 6 within text. 

Response: We thank you for your comment and suggestion.

We have reordered the tables and carefully checked and corrected the order in which they are referenced in the text.

Section 3.2.4.  This whole subsection write-up needs revision for clarification. Long sentences should be broken up. It might prove more efficient to report the statistics in a summary table so that the results can be reported in simpler terms with respect to what the significant findings were. 

Response: We thank you for your comment and suggestion.

We rewrote the subsection, and revised the sentence structure as follows:

There was no significant main effect in the ANOVA results. There were three significant two-way interaction effects, namely, interaction effects between f hemisphere and group, F(2,142)= 3.61, p=0.044, ƞp2 = 0.048, and between hemisphere and emotion regulation(increasing vs decreasing), F(2,142) = 20.47, p<0.001, ƞp2 = 0.224, as well as between hemisphere and anticipated emotion, F(2,142) =40.93, p<0.001, ƞp2 = 0.366. Importantly, there were the significant three-way interaction effects among hemisphere and self-conscious emotion, and emotion regulation, F(2,142) =46.08, p<0.001, ƞp2 = 0.39. There were significant four-way interaction effects between hemisphere and self-conscious emotion, and emotion regulation, and group (HAV vs LAV), F(2,142) =4.79, p=0.015, ƞp2 = 0.063

There was a significant effect between subjects variable of HAV and LAV, F(1,71) = 22.37, p <0.001, Æžp2 = 0.24 (see Figure 7).

Further simple analysis conducted in the HAV environment showed that when the participants received the message of increased anticipated pride, the average amplitude triggered by the emotion regulation was the largest in the left brain area (5.09 ±4.34 µV), and the amplitude of the right side (3.01 ± 4.06 µV) was the smallest. The difference in the average amplitudes between left and right was significant, p < 0.001. The volatility in the middle (3.85 ± 468 µV) was larger than that on the right. In this circumstance, while increased anticipated guilt, the largest amplitude was in the right (3.6 ±4.33 µV) and the smallest was in the left side (3.03 ±4.34 µV). The midline (3.26 ±4.98 µV) was larger than that in the left and there was no significant difference between the three.

Regarding HAV, when the participants received the message of decreased anticipated pride, there was no significant difference in the average amplitudes triggered by the emotion regulation among the hemispheres (see Table 7). In the HAV context, the difference in the lateralization of the left-middle-right brain regions was significant only when anticipated pride was increased. In other emotion regulation situations (increasing anticipated guilt, decreasing anticipated pride, and decreasing anticipated guilt), the difference in lateralization processing was not significant.

Regarding LAV, when the participants received the message of decreased anticipated pride, the average amplitude triggered by the emotion regulation was the largest amplitude of the negative-going in the midline (-0.71 ± 3.36 µV) compared to the left (0.1 ± 2.99 µV) and right (-0.4 ± 3.02 µV) sides. The difference in the average amplitudes between the midline and left was significant (p = 0.016), and the difference between the left and right sides was not significant. When decreasing anticipated pride in the midline (-1.13 ±3.73 µV) was the most negative-going compared to the left (-0.81 ± 2.87 µV) and right (0.05 ± 3.4 µV) side. The difference between the midline and the right side was significant, p<0.001.

The average amplitude while increasing anticipated guilt in the midline (-0.7 ± 3.71 µV) was the most negative-going compared to the left (-0.57 ± 3.02 µV) and right (0.41 ± 3.28 µV) side. The difference in the average amplitudes between the middle and the right side was significant, p<0.001, and the difference between the left and the right side was significant, p = 0.031.

When participants were in the HAV environment, they received the environmental message framing of emotional regulation (increasing or decreasing the anticipated pride), which triggered the average amplitude of LPP changed direction (increasing and decreasing the average amplitude) consistent with the direction of emotional regulation in midline and right areas (see Figure 8). Regarding the LAV environment, the changing direction was the opposite.

Table 7 and Figure 7 suggested ERPs of emotional regulation processing were lateralized to the left and right hemispheres. Specifically, when the participants received the environmental message framing of increasing the anticipated pride, the largest amplitude triggered was generally left-lateralized. When they received the message of decreasing anticipated guilt, it was generally left-lateralized. However, there was a significant difference between the emotion-regulation of self-consciousness for the anticipated pride and guilt in the temporal dynamics of decision-making information processing.

The ERP results showed that in the HAV context, increasing and decreasing anticipated pride corresponded to an increase and a decrease in LPP amplitudes in the left and middle regions, respectively, while increasing and decreasing anticipated pride in the right hemisphere showed the opposite trend. While the trends of increasing and decreasing anticipated guilt in the left and middle brain regions were opposite, only the right side increased. Moreover, decreased anticipated guilt increased and decreased the LPP average amplitude (see Table 7).

The results demonstrated that the emotion regulation adjusted the average amplitude of the LPP and altered their realistic pro-environment behavior.

Figure 7. Why is e included as part of Figure 7? You don't refer to what it represents in the figure caption; seems should be a separate figure on its own?

Response: We thank you for your comment and suggestion.

We separated Figure 7 into two figures, namely Figure 7 and Figure 8, and have included appropriate descriptions in the figure captions

Discussion

Line 463. “Our study explores the effects of mediating anticipatory emotions” You didn’t propose/perform any mediational analysis so it’s unclear as to what you mean here by mediating emotions. 

Response: We thank you for your comment.

We rewrote this sentence as follows: Our study explored the effects of anticipated emotional regulation and aesthetic value environment on pro-environmental behavior and its underlying neural mechanisms.

Line 466. re ‘expected emotions’. There are elsewhere referred to as anticipated emotions – be consistent throughout. 

Response: We thank you for your suggestion.

We have revised the word, and have ensured consistent usage of the term throughout the text.

Lines 469-470. This is an overreaching claim given that you reported some non-significant findings for which we need to accept that there was no difference (i.e., no effect). 

Response: We thank you for your comment and suggestion.

We rewrote the two parts about the non-significant results as follows: The behavioral results suggested that there was no significant difference in the allocation amount to CEPA between HAV and LAV environment groups, p =0.34. Under the circumstance of the interacted effect of emotion regulation and aesthetic value environment, the individual’s pro-environmental behaviors showed no difference regarding HAV and LAV environments.

However, the difference in the pro-environmental behavior between the HAV and LAV environments was not significant. The findings were not consistent with those of previous studies. The reason could be the role of emotion regulation, as we have conducted emotional regulation in HAV and LAV environments and found changes in pro-environmental behavior in the emotion regulation process. The emotional regulation interacted with the perceptions of the HAV and LAV environments, which together regulated an individual’s pro-environmental behavior.

Reference

  1. Wolstenholme, E., W. Poortinga, and L. Whitmarsh, Two birds, one stone: The effectiveness of health and environmental messages to reduce meat consumption and encourage pro-environmental behavioral spillover. Frontiers in psychology, 2020: p. 2596.
  2. Moussaoui, L.S., O. Desrichard, and T.L. Milfont, Do environmental prompts work the same for everyone? A test of environmental attitudes as a moderator. Frontiers in psychology, 2020. 10: p. 3057.

Reviewer 2 Report

The paper is overall a fair work that focuses on a very interesting topic. I advise the authors to address the following comments:

  1. It should be ensured that words included in the title are not iterated as keywords; using different words/phrases in the title and key-words can increase the discoverability of the paper once it is published. 
  2. Please support the sentence in lines 28-30 “Oftentimes, people from different backgrounds may experience the same event in a different manner, resulting in different behaviors” with reference(s).
  3. The sentence “The motivations for protecting the environment are based particularly on … is an individual's pro-environmental behavior” (Lines 33-39) should also be supported with appropriate reference(s).
  4. In lines 82-83, the statement “Therefore, anticipated emotions are particularly relevant for understanding individual decision-making” is not easy to understand and thus needs to be explained and supported.
  5. The sentence in lines 92-94 “Regarding inconsistencies between behavioral decisions and situations, the framing effect is worth mentioning. One of the most robust and important findings in the psychology of decision-making is the dramatic influence of framing” should be referenced. 
  6. The authors need to account for the specific methodology they used in this study. That is, the specific literature sources on which they based their study must be stated.
  7. The study limitations and the recommendations for future studies must be stated more clearly.
  8. In terms of English, the paper is very well-written; however, a careful check for typos should be performed. For instance, in line 336, ‘there’ should be replaced with ‘the’.

Author Response

28 April 2022

Dear expert,

We are writing in reference to our International Journal of Environmental Research and Public Health (IJERPH)submission: “The Influence of Environmental Aesthetic Value and Anticipated Emotion on Pro-environmental Behavior: An ERP Study” [ijerph-1679579].

First of all, we thank you and the two reviewers very much for the recognition of the merits of our revision. We further appreciate your’ valuable feedback and for giving us one more precious opportunity to revise the manuscript. We have dedicated our best efforts to address the comments in this resubmission.

Below, we reproduce the reviewer’s comments, with our responses and changes in the revised manuscript in blue font. Please let us know if there is anything else we could do to improve the manuscript.

Best wishes,

Yiping Zhong,

Professor of Department of Psychology,

Hunan Normal University, Changsha, P.R. China.

Comments and response:

  1. It should be ensured that words included in the title are not iterated as keywords; using different words/phrases in the title and key-words can increase the discoverability of the paper once it is published. 

Response: We thank you for your suggestion, which we strongly agree with. We have modified the keywords as follows: emotional regulation; anticipated guilt; anticipated pride; N1; P2; N2; LPP. We removed the keywords appearing in the title.

  1. Please support the sentence in lines 28-30 “Oftentimes, people from different backgrounds may experience the same event in a different manner, resulting in different behaviors” with reference(s).

Response: We thank you for your comment.

a\It has been reported that people with different social-cultural backgrounds, experiences, and sexes may perceive the same stressors differently;

b\ Due to differences in social background, living environment, learning styles, experiences, and understanding, people pay differing amounts of attention to the same relevant topics and social events, which reflects their heterogeneous characteristics[1, 2]. We have added these references to the text.

  1. The sentence “The motivations for protecting the environment are based particularly on … is an individual's pro-environmental behavior” (Lines 33-39) should also be supported with appropriate reference(s).

Response: We thank you for your comment and suggestion.

a\Under the direction of a series of environmental affection, people generate motivation to protect the environment and tend to engage in pro-environmental behavior[3].

b\Motivation to protect the environment, intrinsic motivation, and self-efficacy also drive favorable environmental behavior[4].

c\ This framework is based on the pro-environmental model developed by Kollmuss and Agyeman (2010), which also asserts the complexity and synergism of internal and external factors in determining an individual’s propensity to partake in pro-environmental behavior that seeks to mitigate the negative impact of their behavior on the environment[5]. We have added these references to the text.

  1. In lines 82-83, the statement “Therefore, anticipated emotions are particularly relevant for understanding individual decision-making” is not easy to understand and thus needs to be explained and supported.

Response: We thank you for your comment and suggestion.

Existing research provides evidence that evoking self-conscious emotions, particularly pride and guilt, in connection with the environmental impact of one’s behaviors, can motivate pro-environmental behavior (PEB) [6-11]. We have added these references to the text.

  1. The sentence in lines 92-94 “Regarding inconsistencies between behavioral decisions and situations, the framing effect is worth mentioning. One of the most robust and important findings in the psychology of decision-making is the dramatic influence of framing” should be referenced. 

Response: We thank you for your comment and suggestion.

Existing evidence suggests that presenting the same option in different manners  alters an individual’s decision[12]. Individuals have a tendency to make different choices according to the presentation of the suggestion (i.e., corresponding to the well-known framing effect)[13]. We have added these references to the text.

  1. The authors need to account for the specific methodology they used in this study. That is, the specific literature sources on which they based their study must be stated.

Response: We thank you for your comment and suggestion.

We stated the specific methodology that was used in this study and added relevant references as follow:

Event-related potentials (ERPs) is a neuroscience technology with a high temporal resolution property, which can be used to investigate the temporal dynamics of decision-making information processing[14-16]

  1. The study limitations and the recommendations for future studies must be stated more clearly.

Response: We thank you for your comment and suggestion. We have revised the Limitations as follows: This study considered only the interaction effects of the HAV and LAV environ-ments and emotion regulation on pro-environmental behavior, and did not explore the influence of the two variables of aesthetic value environment and emotion regulation on pro-environmental behavior, respectively. Future studies should explore the changing footprint of pro-environmental behavior from the perspectives of aesthetic value environment and emotion regulation, and how these variables affect pro-environmental behavior.

  1. In terms of English, the paper is very well-written; however, a careful check for typos should be performed. For instance, in line 336, ‘there’ should be replaced with ‘the’.

Response: We thank you for your comment and suggestion.

We apologize for the typos. In line 336, ‘there’ was replaced with ‘the’. We have checked spelling, grammar, and expressions in the manuscript thoroughly and availed of a professional language editing service to help us proofread the revised manuscript.

Reference

  1. Chen, Y.-Z., et al., Pooled analysis of the Xpert MTB/RIF assay for diagnosing tuberculous meningitis. Bioscience Reports, 2020. 40(1).
  2. Chen, T., et al. Modeling public opinion reversal process with the considerations of external intervention information and individual internal characteristics. in Healthcare. 2020. Multidisciplinary Digital Publishing Institute.
  3. Sheng, G., J. Dai, and H. Pan, Influence of Air Quality on Pro-environmental Behavior of Chinese Residents: From the Perspective of Spatial Distance. Frontiers in Psychology, 2020: p. 2353.
  4. Chen, M., E. Jeronen, and A. Wang, Toward environmental sustainability, health, and equity: how the psychological characteristics of college students are reflected in understanding sustainable development goals. International journal of environmental research and public health, 2021. 18(15): p. 8217.
  5. Eckl, M.R., et al., Replacement of meat with non-meat protein sources: A review of the drivers and inhibitors in developed countries. Nutrients, 2021. 13(10): p. 3602.
  6. Adams, I., K. Hurst, and N.D. Sintov, Experienced guilt, but not pride, mediates the effect of feedback on pro-environmental behavior. Journal of Environmental Psychology, 2020. 71: p. 101476.
  7. Brosch, T. and L. Steg, Leveraging emotion for sustainable action. One Earth, 2021. 4(12): p. 1693-1703.
  8. Ferguson, M.A. and N.R. Branscombe, Collective guilt mediates the effect of beliefs about global warming on willingness to engage in mitigation behavior. Journal of Environmental Psychology, 2010. 30(2): p. 135-142.
  9. Mallett, R.K., K.J. Melchiori, and T. Strickroth, Self-Confrontation via a Carbon Footprint Calculator Increases Guilt and Support for a Proenvironmental Group. Ecopsychology, 2013. 5(1): p. 9-16.
  10. Rees, J.H., S. Klug, and S. Bamberg, Guilty conscience: motivating pro-environmental behavior by inducing negative moral emotions. Climatic Change, 2015. 130(3): p. 439-452.
  11. Hurst, K.F. and N.D. Sintov, Guilt consistently motivates pro-environmental outcomes while pride depends on context. Journal of Environmental Psychology, 2022. 80: p. 101776.
  12. Reyna, V.F., How people make decisions that involve risk: A dual-processes approach. Current directions in psychological science, 2004. 13(2): p. 60-66.
  13. Cassotti, M., et al., Positive Emotional Context Eliminates the Framing Effect in Decision-Making. Emotion, 2012. 12(5): p. 926.
  14. Carlson, R.W., L.B. Aknin, and M. Liotti, When is giving an impulse? An ERP investigation of intuitive prosocial behavior. Social Cognitive and Affective Neuroscience, 2016. 11(7): p. 1121-1129.
  15. Xiao, F., et al., Conflict monitoring and stimulus categorization processes involved in the prosocial attitude implicit association test: Evidence from event-related potentials. Social Neuroscience, 2015. 10(4): p. 408-417.
  16. Li, M., et al., Interpersonal distance modulates the influence of social observation on prosocial behaviour: An event-related potential (ERP) study. International Journal of Psychophysiology, 2022.

Round 2

Reviewer 1 Report

Thanks for the opportunity to review your revised manuscript, which is a great improvement on the original submission. In my opinion, there are additional clarifications needed in places, particularly in the methods and results sections. I've inserted comments in the attached to indicate where I believe further changes are required. 

Author Response

May 4, 2022

Dear Expert:

We are writing in reference to our International Journal of Environmental Research and Public Health (IJERPH) submission, “The Influence of Environmental Aesthetic Value and Anticipated Emotion on Pro-environmental Behavior: An ERP Study.” The manuscript ID is ijerph-1679579.

First of all, we would like to thank you very much for the recognition of the merits of our revision. We further appreciate your valuable feedback and for giving us one more precious opportunity to revise the manuscript. We have dedicated our best efforts to address the comments in this resubmission.

Below, we reproduce the reviewer’s comments with our responses and the changes to the revised manuscript in blue font. Please let us know if there is anything else we can do to improve the manuscript.

Best wishes,

Yiping Zhong,

Professor of Department of Psychology,

Hunan Normal University, Changsha, P.R. China.
